# Data-Informed Geometric Space Selection

**Shuai Zhang**
ETH Zurich
cheungshuai@outlook.com

**Wenqi Jiang**
ETH Zurich
wenqi.jiang@inf.ethz.ch

## Abstract

Geometric representation learning (e.g., hyperbolic and spherical geometry) has proven to be efficacious in solving many intricate machine learning tasks. The fundamental challenge of geometric representation learning lies in aligning the inherent geometric bias with the underlying structure of the data, which is a rarely explored topic in the literature. Existing methods heavily rely on heuristic assumptions on the data structure to decide the type of geometry to be adopted, which often leads to suboptimal performance. This work aims to automate the alignment process via a data-informed strategy such that we optimize model performance with minimal overhead. Specifically, a sparse gating mechanism is employed to enable each input data point $p$ to select $K$ geometric spaces from a given candidate geometric space pool with $N$ ($K < N$) spaces of different geometry. The selected $K$ spaces are then tightly integrated to formulate a Cartesian product space, which is leveraged to process this input data $p$. In doing so, each input data is processed by the spaces it selected with maximum specialization. We empirically show that this method can effectively align data and spaces without human interventions and further boost performance on real-world tasks, demonstrating its potential in eliciting the expressive power of geometric representations and practical usability.

## 1 Introduction

Representing data in low-dimensional geometric spaces (e.g., Euclidean, hyperbolic, and spherical) has been an invaluable technique in a variety of machine learning tasks. Euclidean space has been the most widely adopted and predominant geometric space across many tasks [8, 31]. Recent research has shown that non-Euclidean geometry, such as hyperbolic and spherical geometry, can be more effective in representing data with specific/intricate inherent structures. Typically, hyperbolic space, a type of manifold with constant negative (sectional) curvature, provides a greater ability to model tree-like structures and capture hierarchical patterns [32, 11, 21]. Conversely, spherical space, with positive curvature, is well-suited for fitting data with cyclical structures [44, 30]. Different geometry endows models with different inductive bias that prioritizes learning specific patterns in the data.

Geometric representation learning faces a significant challenge in effectively aligning underlying data structures with appropriate geometric spaces. Earlier approaches tend to map all data to a single geometric space, operating under the implicit assumption that the majority, if not all, of the data conforms to the same geometric pattern. This assumption of a uniform geometric pattern across all data points is unrealistic. As such, there has been growing interest in hybridizing different geometric spaces [22, 37, 4] via Cartesian product to form a mixed curvature space. By leveraging the benefits of multiple geometric spaces, this approach can offer improved performance compared to methods that rely on a single geometric space.

Real-world data is usually characterized by intricate patterns that a single geometric space cannot adequately capture. Thus, employing a one-space-fits-all approach is unlikely to produce optimal solutions. In addition, the Cartesian product space representation learning technique does not discriminate between input examples based on their underlying geometric structures. For example,

37th Conference on Neural Information Processing Systems (NeurIPS 2023).

if a Cartesian product space is constructed by combining a spherical space with a hyperbolic space, both spaces will be used to fit all input data examples, irrespective of whether a data point only aligns with one of the geometry. Ideally, each space should calibrate its internal parameters using the input examples that conform to the corresponding manifold to ensure that the modeling capacity of all geometric spaces is maximized.

Accurately aligning input data with appropriate geometric spaces is a crucial challenge in geometric representation learning. However, neither the single geometric space method nor the Cartesian product approach fully address this issue. Our primary goal is to tackle this challenge by developing an automated approach that seamlessly aligns each data point with its appropriate space. Instead of relying on oversimplified heuristics or manual interventions, we aim to *learn* the data-space alignment component together with the task-specific model using an end-to-end framework.

To this end, we propose a data-informed method for geometric space selection. Given $N$ candidate geometric spaces with different geometry, we let each data point $p$ choose $K(1 \leq K < N)$ spaces from the pool. The chosen $K$ spaces are then tightly combined via Cartesian product to form a Cartesian product space, which is used to process $p$. The selection process is parameterized by a sparsely gated neural network that can adaptively activate a subset of the given spaces based on the given input data. This module can be easily plugged into any geometric representation learning task to enable end-to-end optimization. Theoretically, we can construct $_NC_K = \frac{N!}{(N-K)!K!}$ possible Cartesian product spaces in the end, which brings a greater degree of freedom and personalization for representation learning. Moreover, the proposed method will only incur a negligible computational overhead since only the $K$ selected geometric spaces participate in the model inference no matter how large the space pool is.

The contributions of this work are summarized as follows:

- We propose a data-informed approach for geometric space selection that offers the advantage of automatically aligning each input data point with its preferred geometric space, thereby increasing specificity while being computationally efficient.
- We empirically demonstrate the practical effectiveness of this method with real-world datasets across two downstream tasks. We examine the proposed method's inner workings (e.g., distribution of the selected geometric spaces) via visualization, ablation, and case studies, reaffirming the benefits of the proposed method.

## 2 Related Work

We briefly review two areas of active research that intersect with the present work.

### 2.1 Non-Euclidean Geometric Representation Learning

Two popular non-Euclidean geometric spaces with non-zero curvature, including hyperbolic and spherical spaces, have gained increasing attention and can be found in a broad spectrum of representation learning applications. Hyperbolic space has negative curvature and is reminiscent of a continuous version of trees, which makes it excel in capturing hierarchical structures. A bunch of work has demonstrated its effectiveness in tasks such as graph modeling and knowledge representation [32, 42, 39, 5, 12, 35, 28, 10, 15, 20, 9, 26, 40]. On the other hand, spherical space with positive curvature is more suitable for directional similarity modeling and cyclical-structured data. We can find applications of spherical spaces in text embeddings [30], texture mapping [44], time-warping functions embedding [44]. In these papers, the choice of geometric space is primarily based on expert heuristics about the data characteristics.

To combine the best of different worlds, Gu et al. [22, 37, 4] proposed a method to construct a mixed curvature space via Cartesian product of different spaces. The component spaces in Cartesian product spaces are selected from Euclidean, hyperbolic, and spherical spaces. Each component of Cartesian product space has constant curvature, while the resulting mixed space has a non-constant curvature, which enables it to capture a wide range of patterns with a single model. Empirically, Cartesian product spaces demonstrate its efficacy in graph reconstruction, word embedding with low dimensions [22], node classification in graphs [4], and image reconstruction [37]. Product space is designed to handle data with mixed patterns, but the alignment process between data and geometric

space is still oversimplified and not customized for each data point. Consequently, the challenge of aligning appropriate spaces with input data points persists, and a solution to this problem remains elusive.

### 2.2 Sparsely-gated Mixture of Experts

Another related research area is sparsely-gated neural networks, which are a type of mixture-of-experts (MOE) architecture [24, 25]. Over the past few decades, many MOE architectures have been proposed, such as hierarchical structures [47], sequential experts [2], deep MOE [18], and sparsely-gated MOE [36]. MOE is based on the divide-and-conquer principle, which divides the problem into homogeneous regions and assigns an expert to each region. The final prediction is made by combining the predictions of all experts via a gating network. The gating network controls the contribution of each expert using a probabilistic gating function. If the gating function is sparse, i.e., restricted to assigning non-zero weights to only a subset of the experts, unused experts need not be computed. MOE has been widely studied in various tasks, such as multi-task learning [29], large language models [36, 19], and scaling up vision transformers [34].

We draw inspiration from sparsely gated MOE to automate the alignment between data and its appropriate geometric space. Our approach allows each input data point to control the gating function and select suitable geometric spaces from a candidate geometric space pool, thus enabling customized assignment of data to its suitable spaces. It brings higher specialization, a greater degree of freedom, and superior expressiveness.

## 3 Preliminaries

### 3.1 Riemannian Geometric Space

**Stereographic Projection model.** In general, there are three types of constant curvature spaces with respect to the sign of the curvature (Fig. 1). Common realizations are Euclidean space $\mathbb{E}$ (flat), hypersphere $\mathbb{S}$ (positively curved) and hyperboloid $\mathbb{H}$ (negatively curved). For the latter two, we prefer their stereographic projection model: projected sphere $\mathbb{D}$ and Poincare ball $\mathbb{P}$. These models are easier to optimize and avoid the problem of non-convergence of norm of points with a curvature close to 0, and the projection is conformal, i.e., does not affect the angles between points [32, 37].

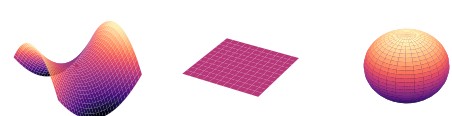

Figure 1: Left: Hyperbolic space $\mathbb{H}$ ; Center: Euclidean plane $\mathbb{E}$ ; Right: Spherical space $\mathbb{S}$ .

An alternative to vector space in non-Euclidean geometry is gyrovector space [41], which defines operations such as vector addition and multiplication.

**Definition 1** *For $\mathbb{D}$ and $\mathbb{P}$ (jointly denoted as $\mathcal{M}_c$, where $c$ denotes curvature), the addition between two points $\mathbf{x}, \mathbf{y} \in \mathcal{M}_c$, also known as Möbius addition $\oplus_c$ (for both signs of $c$), is defined as:*

$$\mathbf{x} \oplus_c \mathbf{y} = \frac{(1 - 2c\langle \mathbf{x}, \mathbf{y} \rangle - c\|\mathbf{y}\|_2^2)\mathbf{x} + (1 + c\|\mathbf{x}\|_2^2)\mathbf{y}}{1 - 2c\langle \mathbf{x}, \mathbf{y} \rangle + c^2\|\mathbf{x}\|_2^2\|\mathbf{y}\|_2^2}, \tag{1}$$

*Where $\langle , \rangle$ is Euclidean inner product. The distance between points in the gyrovector space is defined as:*

$$d_{\mathcal{M}_c}(\mathbf{x}, \mathbf{y}) = \frac{2}{\sqrt{|c|}} \tan_c^{-1}(\sqrt{|c|}\| - \mathbf{x} \oplus_c \mathbf{y}\|_2), \tag{2}$$

*where $\tan_c$ stands for $\tan$ if $c > 0$ and $\tanh$ if $c < 0$. In both spaces, we have Euclidean geometry when $c \to 0$. It is easy to prove that:*

$$d_{\mathcal{M}_c}(\mathbf{x}, \mathbf{y}) \xrightarrow{c \to 0} 2\|\mathbf{x} - \mathbf{y}\|_2, \tag{3}$$

*which means the gyrospace distance converges to Euclidean distance when limiting $c$ to zero.*

**Definition 2** *Let $\mathcal{T}_{\mathbf{x}}\mathcal{M}_c$ be the tangent space to the point $\mathbf{x} \in \mathcal{M}_c$. Mapping between (Euclidean) tangent space and hyperbolic/spherical space is performed with exponential map: $\mathcal{T}_{\mathbf{x}}\mathcal{M}_c \to \mathcal{M}_c$ and logarithmic map: $\mathcal{M}_c \to \mathcal{T}_{\mathbf{x}}\mathcal{M}_c$, which are defined as:*

$$
\begin{aligned}
\log_{\mathbf{x}}^c(\mathbf{y}) &= \frac{2}{\sqrt{|c|}\lambda_{\mathbf{x}}^c} \tan_c^{-1}(\sqrt{|c|}\| - \mathbf{x} \oplus_c \mathbf{y}\|_2) \frac{-\mathbf{x} \oplus_c \mathbf{y}}{\| - \mathbf{x} \oplus_c \mathbf{y}\|_2}, \\
\exp_{\mathbf{x}}^c(\mathbf{v}) &= \mathbf{x} \oplus_c (\tan_c(\sqrt{|c|}\frac{\lambda_{\mathbf{x}}^c\|\mathbf{v}\|_2}{2}) \frac{\mathbf{v}}{\sqrt{|c|}\|\mathbf{v}\|_2}),
\end{aligned}
\tag{4}
$$

*where $\lambda_{\mathbf{x}}^c$ is a conformal factor, defined as $\lambda_{\mathbf{x}}^c = 2/(1 + c\|\mathbf{x}\|_2^2)$, which is used to transform the metric tensors between (Euclidean) tangent space and non-Euclidean space.*

### 3.2 Cartesian Product of Spaces

**Definition 3** *Product space is defined as the Cartesian product of multiple spaces with varying dimensionality and curvature. Let $\mathcal{P}$ denote a product space composed by $N$ independent component spaces $\mathcal{M}^{(1)}, \mathcal{M}^{(2)},...,\mathcal{M}^{(N)}$. The mixed space $\mathcal{P}$ has the form (operator $\times$ can be omitted):*

$$
\mathcal{P} = \bigtimes_{i=1}^{N} \mathcal{M}^{(i)} = \mathcal{M}^{(1)} \times \mathcal{M}^{(2)} \times ... \times \mathcal{M}^{(N)}.
\tag{5}
$$

**Definition 4** *The product space $\mathcal{P}$ also has distance functions. The squared distance between points $\mathbf{x}, \mathbf{y} \in \mathcal{P}$ is defined as:*

$$
d_{\mathcal{P}}^2(\mathbf{x}, \mathbf{y}) = \sum_{i=1}^{N} d_{\mathcal{M}^{(i)}}^2(\mathbf{x}_{\mathcal{M}^{(i)}}, \mathbf{y}_{\mathcal{M}^{(i)}}),
\tag{6}
$$

*where $\mathbf{x}_{\mathcal{M}^{(i)}}$ and $\mathbf{y}_{\mathcal{M}^{(i)}}$ denote the corresponding vectors on the component space $\mathcal{M}^{(i)}$.*

Other operations such as exponential map and logarithmic map are element-wise, meaning that we can decompose the points into component spaces, apply operations on each component space, and then compose them back (e.g., concatenation) to the product space. A product space's signature (i.e., parametrization) refers to the types of space, the number of spaces, the dimensionality, and the curvature of each space. For instance, $(\mathbb{P}^{100})^3(\mathbb{D}^{50})^2$ denotes a Cartesian product space of three hyperbolic spaces of hidden dimension 100 and two spherical spaces of hidden dimension 50.

## 4 Data-Informed Space Selection

### 4.1 Geometric Space Pool

Suppose we have a space pool with $N$ geometric spaces, $\mathcal{M}_c^{(i)} \in \{\mathbb{E}, \mathbb{D}, \mathbb{P}\}, i = \{1, ..., N\}$. For simplicity, we assume all spaces have the same dimensionality $b$. The goal is to select $K(1 \leq K < N)$ spaces from the given $N$ spaces for each data point $p$. If $K > 1$, the selected $K$ spaces will form a product space. Theoretically, there will be $_NC_K = \frac{N!}{(N-K)!K!}$ possible space combinations. Figure 2 illustrates the proposed data-informed space selection pipeline. Here, $\mathbb{P}$ and $\mathbb{E}$ are selected by data $p$ and the Cartesian product $\mathbb{P} \times \mathbb{E}$ is used to handle $p$.

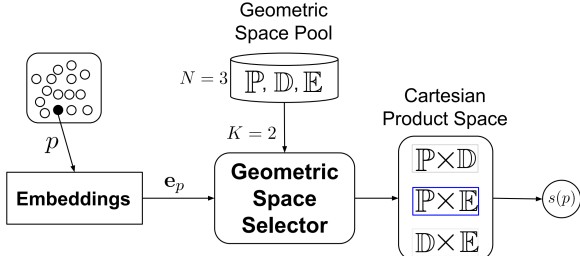

Figure 2: Data-informed geometric space selection. $\mathbb{P}$ and $\mathbb{E}$ are selected to form $\mathbb{P} \times \mathbb{E}$ for input data $p$.

### 4.2 Embeddings

For each geometric space $\mathcal{M}_c^{(i)}$, we define an embedding dictionary $\mathbf{e}^{(i)}$. In the $i^{\text{th}}$ space, data point $p$ is represented by a vector $\mathbf{e}_p^{(i)} \in \mathbb{R}^b$, where $b$ is the hidden dimension of this embedding dictionary.

We initialize all the embedding vectors in the tangent space, and the exponential map will be applied to recover them into hyperbolic or spherical space when necessary. In doing so, standard Euclidean optimization algorithms can be directly applied to enable an end-to-end learning framework to avoid the cumbersome Riemannian optimization [7].

## 4.3 Geometric Space Selector via Sparsely-gated MOE

We use a sparsely-gate MOE network to select a suitable space subset from the space pool. The selector is a differentiable network that takes as input the embedding vectors $\mathbf{e}_p^{(i)}$ and outputs the probability of each space to be selected. Formally, the input is defined as:

$$\mathbf{x}_p = [\mathbf{e}_p^{(1)}, \mathbf{e}_p^{(2)}, ..., \mathbf{e}_p^{(N)}]^\top, \tag{7}$$

where $\mathbf{x}_p$ is a two-dimensional matrix of shape $N \times b$. We use a convolutional neural network to transform this matrix into a vector of dimension $N$. Other types of neural networks are also viable, we adopt CNN because it does not introduce a lot of additional parameters. Then, the last hidden layer of the gating network has the following form:

$$f(\mathbf{x}_p) = f_1(\mathbf{x}_p) + \text{randn}() \cdot \ln(1 + \exp(f_2(\mathbf{x}_p))), \tag{8}$$

where $f_1$ and $f_2$ represent CNN layers and $f_*(\mathbf{x}_p) \in \mathbf{R}^N$; function "randn()" is used to product Gaussian noise in the training stage to improve loading balancing, i.e., balance the number of samples accepted by each space.

To impose sparsity, we employ a TopK function to obtain the largest $K$ elements from $f(\mathbf{x}_p)$:

$$f(\mathbf{x}_p) \leftarrow \text{TopK}(f(\mathbf{x}_p)), \tag{9}$$

where TopK returns the original value if the element is in the top $K$ list; otherwise, it returns $-\infty$. Afterward, $N$ softmax gating functions $g_1, g_2, .., g_N$ are leveraged to control which $K$ space(s) to be selected. For $-\infty$, it will output a zero.

$$g_i(\mathbf{x}_p) = \frac{\exp(f(\mathbf{x}_p)_i)}{\sum_{j=1}^N \exp(f(\mathbf{x}_p)_j)}, i = \{1, ..., N\}. \tag{10}$$

Later, spaces with nonzero gating values will be activated while others remain idle. The selected $K$ space will form a Cartesian product space.

## 4.4 Task Specific Prediction Function

The prediction function is specific to the type of task. For instance, in each component space $\mathcal{M}_c^{(i)}$, we can adopt squared distance to produce the model prediction. Given two points $p$ and $q$, the prediction function can be defined as $s_{\mathcal{M}_c^{(i)}}(\mathbf{e}_p^{(i)}, \mathbf{e}_q^{(i)})$. If space $i$ is not selected, $s_{\mathcal{M}_c^{(i)}}$ is set to 0. More details will be provided in the experiment section for each task. If $K > 1$, the final prediction score is computed as:

$$s(p, q) = \log(\sum_{i=1}^N g_i(\mathbf{x}_p) \exp(s_{\mathcal{M}^{(i)}}(\mathbf{e}_p^{(i)}, \mathbf{e}_q^{(i)}))). \tag{11}$$

Here, a log-sum-exp technique is leveraged to improve the numerical stability to avoid problems such as underflow and overflow. Multiplying the prediction by the gating probability is optional in the framework.

If $s_{\mathcal{M}^{(i)}}$ is calculated with squared distance and $K = N$, we can recover the squared distance function defined for product space by removing $g_i(\mathbf{x})$ and the log-sum-exp technique. Our prediction function cannot be viewed as a standard distance metric as it does not satisfy the triangle inequality property. Also, it is flexible to use other prediction functions to replace squared distances.

## 4.5 Load Balancing Regularization

We use a load balancing regularization term to improve numerical stability and avoid some spaces from being over-selected. In specific, two additional regularization terms are used following [36].

For a batch of $X$ inputs,

$$\ell_1 = \psi(\sum_{p \in X} g(\mathbf{x}_p))^2, \tag{12}$$

where $\psi(v) = \frac{variance(v)}{mean(v)}$ is the coefficient of variation.

$$\ell_2 = \psi(\kappa(X))^2, \tag{13}$$

$$\kappa(X)_i = \sum_{p \in X} \Phi(\frac{f_1(\mathbf{x}_p) - \xi(f(\mathbf{x}_p), k, i)}{\ln(1 + \exp(f_2(\mathbf{x}_p)))})), \tag{14}$$

where $\xi(v, k, i)$ is the $k^{\text{th}}$ highest component of $v$ excluding component $i$. $\Phi$ is the cumulative distribution function of standard normal distribution.

The two regularization terms are added to the task-specific loss for model optimization:

$$\ell = \ell_{\text{task}} + \mu_1 \ell_1 + \mu_2 \ell_2, \tag{15}$$

where $\mu_1$ and $\mu_2$ are scaling factors (set to $0.01$ by default).

## 4.6 Limitation

Compared with Cartesian product space [22], the proposed method introduces one additional hyperparameter, $K$. As we will show later, a relatively small $K$ (e.g., $K <= 5$) can introduce a significant performance boost. Just like determining the signature of Cartesian product space, we need to determine what geometric spaces to be included in the space pool. This overhead is inherited from Cartesian product space and can be potentially mitigated via hyper-parameters optimization algorithms [23].

## 5 Experiments

In this section, we evaluate the proposed method on real-world datasets to demonstrate its capability in dealing with practical tasks, including personalized ranking and link prediction for relational graphs.

### 5.1 Personalized Ranking

The task of personalized ranking is to provide a user with a tailored, ranked list of recommended items [33]. Given a partially-observed interaction matrix between users and items, where each entry of the matrix represents the rating/preference a user $u$ gives to an item $v$ if $u$ has interacted with the iterm (e.g., watched the movie) and is otherwise missing. We conduct experiments on two well-known datasets, MovieLens 100K and MovieLens 1M. The first dataset consists of 100K ratings between 943 users and 1,682 movies, and the second dataset consists of one million ratings between 6,040 users and 3,706 movies. Over 90% of the entries of these interaction matrices are missing. Completing this matrix enables us to recommend to customers what to watch next.

The personalized ranking model takes a user vector $\mathbf{u}_u^{(i)} \in \mathbb{R}^b$ and an item vector $\mathbf{v}_v^{(i)} \in \mathbb{R}^b$ as input, and approximate the rating with squared distance [42]. That is, the rating of user $u$ gives to movie $v$ is estimated by:

$$s_{\mathcal{M}^{(i)}}(u, v) = -d^2_{\mathcal{M}^{(i)}}(\exp_{\mathbf{0}}^c(\mathbf{u}_u^{(i)}), \exp_{\mathbf{0}}^c(\mathbf{v}_v^{(i)})). \tag{16}$$

Alternatively, we can use the dot product, $\mathbf{u}_u^{(i)} \cdot \mathbf{u}_v^{(i)}$, to approximate each entry of the matrix [33].

The input of the gating network is the concatenation of the user and item embeddings. Specifically, a vector of size $\mathbb{R}^{2Nb}$ is used as the input, and a linear layer is used as the sparsely-gated network. Let $s(u, v)$ denote the final prediction. We treat the task as a learning-to-rank problem and optimize it by minimizing the following contrastive loss:

$$\ell_{\text{task}} = \sum_{(u, v_1, v_2) \in \Omega} \max(0, s(u, v_1) + m - s(u, v_2)), \tag{17}$$

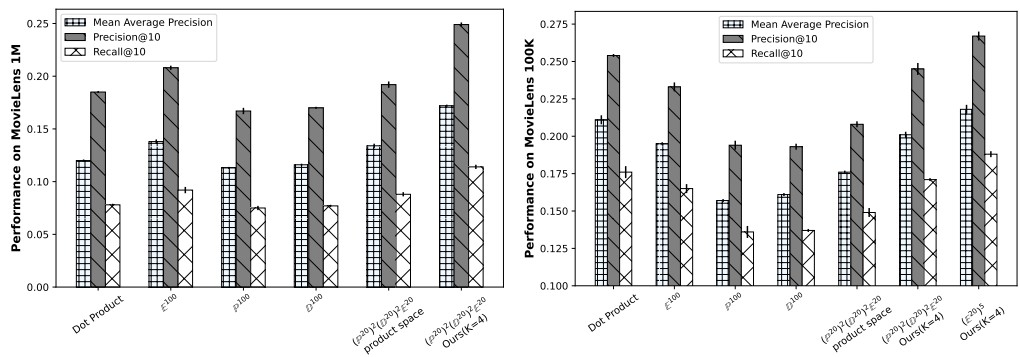

Figure 3: Personalized ranking performance on MovieLens 1M and MovieLens 100K.

where $\Omega$ is the collection of training samples, $v_1$ is the movie that $u$ watched, and $v_2$ represents an unobserved movie for the user; $m$ is a hyper-parameter.

We hold 70% entries in each user's interactions as the training set, 10% entries as the validation set for model tuning, and the remaining 20% for model testing. All interactions (e.g., ratings) are binarized following the implicit feedback setting [33]. For all methods, the total dimension $\sum_{i=1}^{N} b_i$ is set to 100 for a fair comparison to ensure the same model size. The curvatures for spherical and hyperbolic models are set to 1 and $-1$, respectively. $N$ is set to 5. $K$ is tuned among $\{1, 2, 3, 4\}$. Regularization rate is chosen from $\{0.1, 0.01, 0.001\}$. $m$ is fixed to 0.5. Adam is adopted as the optimizer. We measure the performance based on the widely adopted metrics in personalized ranking: mean average precision (MAP), precision, and recall at the top-ranked ten items (Precision@10 and Recall@10). For all experiments, we report the average over five runs.

**Main Results.** Figure 3 summarizes the performances. The proposed method achieves the best scores on all three metrics, outperforming the product space counterpart with the same signature and other single space models. Also, dot-product based approach remains competitive in this task. We estimate the global average curvature with the algorithm described in [22] and obtain a value of 0.695 for MovieLens 1M and 0.19 for MovieLens 100K, which suggests that they lean towards cyclical structures. However, from the performance comparison among $\mathbb{E}^{100}$, $\mathbb{D}^{100}$, and $\mathbb{P}^{100}$, we find that the model built in Euclidean space offers the best performance, outperforming spherical and hyperbolic counterparts. This observation indicates that the global average curvature estimation algorithms proposed by Gu et al. [22] may give misleading information, which also ascertains the difficulty of data-space alignment.

## 5.2 Link Prediction for Relational Graphs

Relational graphs have emerged as an effective way to integrate disparate data sources and model underlying relationships. Encoding the nodes and relations of relational graphs into low-dimensional vector spaces is vital to downstream applications such as missing facts completion, question answering, information extraction, and logical reasoning [3].

Given a relational graph $\mathcal{G}$ with a set of entities $\mathcal{E}$ and a set of relations $\mathcal{R}$. Each triple, abbreviated as $(h, r, t)$, in $\mathcal{G}$ is composed by two entities (i.e., head entity $h \in \mathcal{E}$ and tail entity $t \in \mathcal{E}$), and the relationship $r \in \mathcal{R}$ between them. In the $i^{\text{th}}$ space, entities $h, t$ are represented by vectors $\mathbf{e}_h^{(i)}, \mathbf{e}_t^{(i)} \in \mathbb{R}^b$ and relation $r$ is represented by two translation vectors $\boldsymbol{\alpha}_r^{(i)}, \boldsymbol{\beta}_r^{(i)} \in \mathbb{R}^b$ and a rotation vector $\boldsymbol{\gamma}_r^{(i)} \in \mathbb{R}^b$. Also, each head (tail) entity is associated with a bias term $b_h(b_t) \in \mathbb{R}$.

Inspired by the method (RotE and RotH) proposed in [11], we propose a generic model, RotX, that subsumes the two methods and can be extended to spherical space. In RotX, the head entity is translated twice via Möbius addition and rotated once. Formally, the head entity is processed as follows:

$$Q^{(i)}(h, r) = \text{ROTATE}(\exp_{\mathbf{0}}^c(\mathbf{e}_h^{(i)}) \oplus_c \exp_{\mathbf{0}}^c(\boldsymbol{\alpha}_r^{(i)}), \boldsymbol{\gamma}_r^{(i)}) \oplus_c \exp_{\mathbf{0}}^c(\boldsymbol{\beta}_r^{(i)}), \qquad (18)$$

where $\exp_{\mathbf{0}}^c$ is the exponential map over origin. ROTATE is a rotation function, and $\boldsymbol{\gamma}_r^{(i)}$ is the rotation matrix. The transformed head entity is then compared with the tail entities using squared distance.

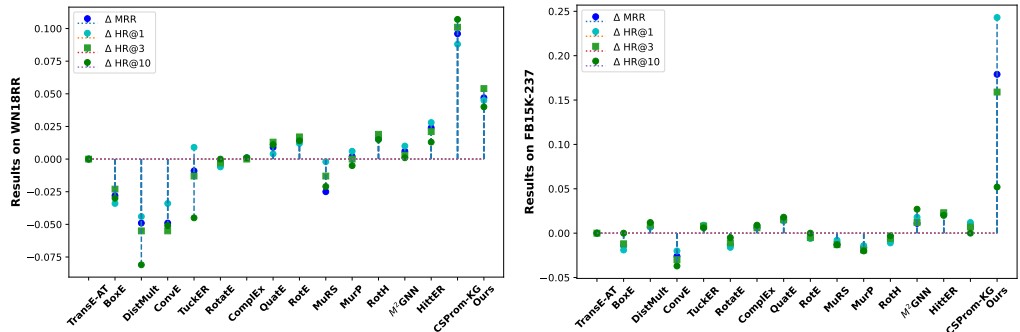

Figure 4: Performance on relational graphs link prediction. We use the performance of TransE-AT [46] as the baseline (e.g., $\Delta$MRR = MRR(Model) - MRR(TransE-AT)) for the lollipop charts. In our method, for WN18RR, the geometric space pool consists of $4 \times \mathbb{D}^{100}$ and one $\mathbb{E}^{100}$, $K = 4$; for FB15K-237, the geometric space pool consists of $3 \times \mathbb{P}^{100}$ and $2 \times \mathbb{D}^{100}$, $K = 4$.

The final scoring function is as follows:

$$s_{\mathcal{M}^{(i)}}(h, r, t) = -d^2_{\mathcal{M}^{(i)}}(Q^{(i)}(h, r), \exp^c_{\mathbf{0}}(\mathbf{e}^{(i)}_t)) + b^{(i)}_h + b^{(i)}_t. \tag{19}$$

We make the curvature relation specific and trainable. Each relation has a curvature parameter and is trained simultaneously with the model. We adopt the following cross-entropy loss as the objective function:

$$\ell_{\text{task}} = \sum_{(h,r,t) \in \Omega} \log(1 + \exp(-Y_{(h,r,t)}s(h, r, t))), \tag{20}$$

where $Y_{(h,r,t)} \in \{1, -1\}$ is a binary label indicating whether a triple is factual (1) or not (-1). $\Omega$ represents the training collection, including positive and negative triples.

We use two datasets including WN18RR [8, 16] and FB15K-237 [8, 16] for model evaluation. WN18RR is taken from WordNet, a lexical database of semantic relations between words. It has $40, 943$ entities, 11 relations, and $86, 835/3, 034/3, 134$ training/validation/test triples. FB15K-237 is a subset of the Freebase knowledge graph, a global resource of common and general information. It has $14, 541$ entities, 237 relations, and $272, 115/17, 535/20, 466$ training/validation/test triples. The performance is evaluated using two standard metrics, including mean reciprocal rank (MRR) and hit rate (HR) with a given cut-off value $\{1, 3, 10\}$.

The total dimension is fixed to $500$ for a fair comparison. Learning rate is tuned amongst $\{0.01, 0.005, 0.001\}$. For all experiments, we report the average over 5 runs. We set the kernel size to 5 and stride to 3 for convolution operation in the gating network. $N$ is set to 5 and $K$ is tuned among $\{1, 2, 3, 4\}$. The number of negative samples (uniformly sampled) per factual triple is set to 50. Optimizer Adam is used for model learning. We perform early stopping if the validation MRR stops increasing after 10 epochs.

We compare our method with several baselines, including Euclidean methods TransE [8], Dist-Mult [45], ConvE [16], TuckER [6], RotE [11], BoxE [1]; complex number based methods ComplEx-N3, [27] and RotatE [38]; quaternion model QuatE [48]; spherical models MuRS [43]; hyperbolic methods MurP [5], RotH [11]; Cartesian product space model $M^2$GNN [43], and Transformer/pretrained model based models HittER [14] and CSProm-KG [13][1]. CSProm-KG uses the pretrained large language model (LLM), BERT [17], as the backbone.

**Main Results.** We report the performance comparison in Figure 4 where we use TransE-AT [46] as baseline[2] and report the differences. We make the following observations. Firstly, our method outperforms all compared baselines without pretrained large language model as external sources. Specifically, it outperforms Euclidean, hyperbolic, spherical, complex-valued, and Cartesian product spaces methods and obtains a clear performance gain over the second-best model. Secondly, the

---

[1]A concurrent work.

[2]TransE-AT on WN18RR: MRR=0.479, HR@1=0.434, HR@3=0.495, HR@10=0.571; on FB15k-237: MRR=0.351, HR@1=0.257, HR@3=0.386, HR@10=0.538.

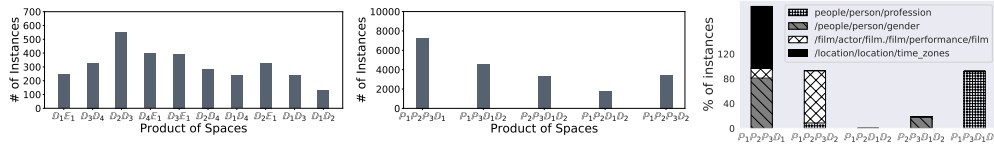

Figure 5: Left: distribution of the selected geometric space on WN18RR; Center: distribution of the selected geometric space on FB15K-237; Right: geometric space distribution of four relations from FB15K-237.

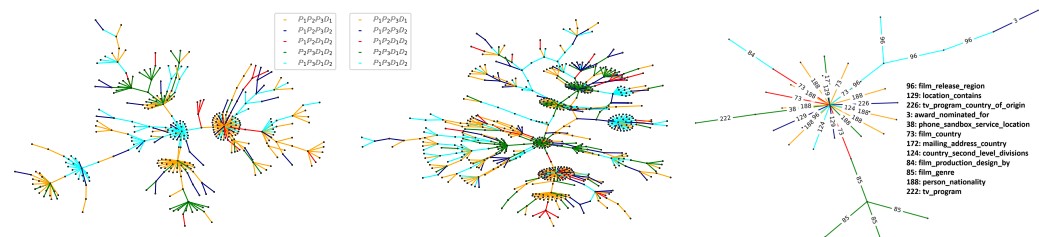

Figure 6: Distribution of the selected spaces shown in a sub-graph of the FB15K-237.

improvement over the second-best model on FB15K-237 is larger than that on WN18RR. One possible explanation is that FB15K-237 has a more diverse set of relations and a higher structure heterogeneity [5], so a better geometric space alignment could bring more value. It is worth noting that WN18RR is sampled from the WordNet lexical database and BERT is trained on a large corpus of lexical data, it is unsurprising that CSProm-KG can obtain very high score on WN18RR. However, its relatively modest scores on FB15K-237 suggest limited practicality for non-lexical knowledge graphs. As such, it is not fair to compare an LLM based approach with our approach as using LLM may lead to data leakage.

**Distribution of the selected Geometric Space.** Figure 5 (left and center) presents the distribution of the selected spaces. We observe that certain product spaces are preferred depending on the dataset. For example, the product space $\mathbb{D}_2\mathbb{D}_3$ is the most selected space on WN18RR. While on FB15K-237, the product space $\mathbb{P}_1\mathbb{P}_2\mathbb{P}_3\mathbb{D}_1$ is more preferable. To show more fine-grained examples, we randomly select four relations from FB15K-237 and visualize the distribution in Figure 5 (Right). We find that each relation has their own desirable product spaces. For example, relation "*/location/ location/time_zones*" prefers space $\mathbb{P}_1\mathbb{P}_2\mathbb{P}_3\mathbb{D}_1$ but relation "*people/person/profession*" prefers space $\mathbb{P}_1\mathbb{P}_3\mathbb{D}_1\mathbb{D}_2$. This reconfirms the specification capability of our method.

To show how the space distribution coincides with data structures, we randomly extract three connected graph components from the test set of FB15K-237 and visualize them in Figure 6. Each triple can be represented with an edge and its two end nodes. We render edges with different colors based on the type of product spaces used to model the corresponding triples. From the left two graphs, we see some clear clusters with a dominant color, which suggests that neighborhood triples are more likely to be processed with the same selected product space. From the right-most graph, we find that different relations usually prefer different spaces, which indicates that the type of relations plays a critical role in the space selection process.

| $\mathcal{M}$ | Signature | WN18RR | | FB15K-237 | |
|---|---|---|---|---|---|
| | | MRR | HR@3 | MRR | HR@3 |
| Single | $\mathbb{D}^{500}$ | 0.492 | 0.514 | 0.293 | 0.322 |
| Product | $(\mathbb{P}^{100})^3(\mathbb{D}^{100})^2$ | 0.484 | 0.498 | 0.311 | 0.344 |
| | $(\mathbb{D}^{100})^4\mathbb{E}^{100}$ | 0.479 | 0.497 | 0.312 | 0.344 |
| | $(\mathbb{P}^{100})^4\mathbb{D}^{100}$ | 0.468 | 0.488 | 0.321 | 0.356 |
| | $(\mathbb{P}^{100})^2(\mathbb{D}^{100})^2\mathbb{E}^{100}$ | 0.479 | 0.496 | 0.308 | 0.342 |
| Ours | $3 \times \mathbb{P}^{100}, 2 \times \mathbb{D}^{100}$ | 0.500 | 0.521 | 0.530 | 0.545 |
| | $4 \times \mathbb{D}^{100}, \mathbb{E}^{100}$ | 0.526 | 0.549 | 0.525 | 0.531 |
| | $4 \times \mathbb{P}^{100}, \mathbb{D}^{100}$ | 0.504 | 0.526 | 0.515 | 0.527 |
| | $2 \times \mathbb{P}^{100}, 2 \times \mathbb{D}^{100}, \mathbb{E}^{100}$ | 0.522 | 0.546 | 0.526 | 0.535 |

Table 1: Comparison with Cartesian product space models [22] (a few signatures are reported due to length constraints). $K = 2$ for WN18RR and $K = 4$ for FB15K-237.

prefer different spaces, which indicates that the type of relations plays a critical role in the space selection process.

**Comparison with Cartesian Product Space with Varying Signatures.** Compared with M²GNN (its signature is $\mathbb{P}^{200}\mathbb{D}^{200}\mathbb{E}^{200}$) (in Figure 4). Our model achieves the best performance even with

fewer trainable parameters. Then, we compare our method with Cartesian product space models with varying signatures in Table 1. We observe that our method constantly outperforms pure product spaces with the same prior signatures. It is worth noting that our method requires merely two/four active spaces to outperform product space models with five active spaces. These observations reconfirm the effectiveness of the data-informed space alignment method.

**Impact of N and K.**

The left figure of 7 shows the impact of $N$ by fixing $K$. We observe that (a) the model performance does not benefit much from increasing $N$ *on this dataset*; (b) the inference time remains nearly constant when we increase $N$, confirming its efficiency. The effect of $K$ is shown in Figure 7 (right). We find that $K$ has a higher impact on the

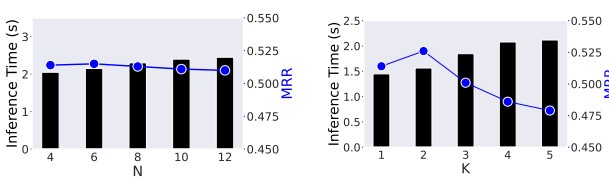

Figure 7: Effect of $N$ (left) and $K$ (right) on WN18RR.

model performance and inference time. Increasing $K$ will generally degrade the performance *on this dataset* and yield additional computational overhead.

## 6 Conclusion

In this paper, we propose a data-informed geometric space selection method. Our method enables an automatic alignment between data and geometric spaces by allowing each data point to automatically select its preferred geometric spaces via a sparsely-gated MOE network. It brings a greater extent of specification for geometric representations while remaining efficient. We show that the proposed approach can boost representation performance in real-world applications with a noticeable margin.

**Broader Impact.** This approach exhibits considerable potential in facilitating enhanced and versatile geometric representation learning, thereby engendering broader and more comprehensive applications across various domains.

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
