# OpenReview forum: "Data-Informed Geometric Space Selection"
_NeurIPS.cc/2023/Conference — NeurIPS 2023 poster_

### Official Review · Reviewer_c7Xk · 2023-07-02

**Soundness:** 3 good
**Presentation:** 2 fair
**Contribution:** 2 fair
**Rating:** 5
**Confidence:** 4

**Summary:**

The paper proposes a new end-to-end distance learning method to facilitate solving downstream prediction tasks. Their core idea is to select a subset of geometric spaces from a candidate space sets that include Euclidean, projected sphere and Poincare ball types of spaces, and computes the final distance in the Cartesian product space of the selected spaces.  The selector is built by applying the sparsely-gated MOE technique. To improve the stability of the training, some balancing regularization terms are added to the training loss. Performance of the proposed method is assessed by one matrix completion task and one link prediction task using benchmark datasets. Reported results show performance improvement.

**Strengths:**

The proposed idea has some originality (a kind of creative combination of existing ideas) and the proposed algorithm design is reasonable.

**Weaknesses:**

A major weakness is the writing and clarity. First of all, Sections 3.1 and 3.2 are not proposed method, but a short repeat of Section 2 of [34], but in poorer quality. The subsection of “Geometric Space Selector via Sparsely-gated MOE” from line 172 to line 188 is almost a repeat of Section 2.1 of [33] published in ICLR 2017. The design motivations of the two balancing regularisation terms are not clearly explained, especially for l_2 that is not as straightforward as l_1. It is not clear that how to construct different candidate geometric spaces. The authors only mention that there are three types of spaces. How exactly to compute distance using tangent vectors in a geometric space is not explained. More such information in the Embedding section in line 166 would be helpful. It is a pity that in the proposed method section the authors use quite some space for a poor summary of existing knowledge rather than putting good effort to explain well their own proposed method.

It seems that the goal of the research is to boost prediction performance by improving geometric representation learning. Under such goal, experiment section has weakness. It only compares between different spaces, but lacks comparison with published results by mainstream existing works and state of the art results on the same benchmark datasets for both matrix completion and link prediction. Without such comparison, it is hard to recognise the value/importance of  geometric representation learning for solving downstream prediction tasks. Ablation studies on the effect of the regularization terms are missing.

The contribution is a little limited given that the only objective seems to boost performance. Under such goal, the direction of geometric representation learning may not even be the best direction to pursue for each prediction task. It would be more interesting if the work could explore other potential, in addition to prediction performance improvement, of geometric representation learning.

**Questions:**

(1)	Explain how the two candidate hyperbolic and two spherical spaces were constructed?

(2)	In line 236, the authors said that the training examples are formed based on watched (v1) and unobserved (v2) movies. This is odd. In MovieLens data, the matrix elements are ratings like 1,2,3,4,5.  Isn’t the matrix completion supposed to predict the ratings of unobserved movies for a user?  Why is it meaningful to push s(u,v1) -s(u,v2) +m < = 0?

(3)	Can the authors perform a brief investigate on what experiment settings some state-of-the-art works on matrix completion and link prediction have used for the same three benchmark datasets, and what performance they have obtained, and compare the published results with the authors' results?

(4)	Perform some ablation studies to examine the role of l_1 and l_2.

**Limitations:**

Discussion on geometric representation learning is quite basic. It would be interesting to discuss why it is important and useful to research geometric representation learning, and whether the proposed approach has other potentials, in addition to boosting prediction performance.

---

> ### Author Rebuttal · Authors · 2023-08-10
>
> Thank you very much for the review, especially the constructive suggestion on the writing/clarification. We will revise the manuscript as suggested and clarify some of the confusing points.
>
> ##### **For Section 3.1 and section 3.2**
> We will move Section 3.1 and Section 3.2  to a separate preliminary section to ensure the paper to be self-contained.
>
> ##### **The design motivations of the two balancing regularisation terms are not clearly explained**
> - In the first regularization loss (eq-12), the batch-wise sum of the gate values are considered as the importance of an expert. Minimizing the coefficient of variation the importance distribution encourages all experts to have equal importance.
> - The second loss (eq-13) is used to encourage all experts to receive roughly equal numbers of training examples. Although one can ensure equal expert importance with the first loss, the number of training examples received by each expert can be different (e.g., one expert receives a few examples with large weights, and the other receives many examples with small weights). Here, $\kappa$ can be viewed as a soft estimator of the number of examples assigned to each expert for a batch of inputs. Minimizing the coefficient of variation this distribution will help encourage an equal number of training examples for each expert.
>
> ##### **Explain how the two candidate hyperbolic and two spherical spaces were constructed?**
> We will provide a formal description in the paper (e.g., after line 166).
>
> Taking the recommendation task as an example, suppose that we have two candidate hyperbolic spaces ($P_1$, $P_2$)  and two spherical spaces ($D_1$, $D_2$) in the candidate space pool.
> In the first step, we will initialize four embedding vectors for each item (say $ i_1, i_2, i_3, i_4$) and four embedding vectors for each user ($u_1, u_2, u_3, u_4$), and all vectors are in the tangent space.
> Secondly, we assume that $i_1$ and $u_1$ are in the hyperbolic space $P_1$; $i_2$ and $u_2$ are in the hyperbolic space $P_2$; $i_3$ and $u_3$ are in the spherical space $D_1$; $i_4$ and $u_4$ are in the same spherical space $D_2$. The distance of $i_*$ and $u_*$ is computed via eq(16).
> Thirdly, if we set $K=2$, and the model selected $P_1$ and $D_2$ for the final prediction. We will combine the distance in $P_1$ (computed by $i_1$ and $u_1$) and distance in $D_2$ (computed by $i_4$ and $u_4$) via equation (11).
>
>
> ##### **How exactly to compute distance using tangent vectors*
> Because we have two separate applications, we the distance for each application is defined in different sections. One can get the final distance calculation from eq 2, 11, 16 (for matrix factorization) and 19 (relational link prediction)
>
>
> ##### **Loss for Movielens-1m**
> For MovieLens 1M, as we mentioned in line 240, we have binarized the ratings into 0/1, where ratings greater than or equal to 4 are treated as 1, otherwise, it is treated as unknown (0). This is a common setting in matrix factorization/collaborative filtering. As in most real-world applications, feedback is in the form of likes/dislikes, viewed/not viewed, etc. As such, we adopted this contrastive loss function.
> To avoid confusion, we will rename it to **matrix factorization** instead of matrix completion.
>
> ##### **Compare with STOA**
>
> We provide comparisons with other stoa methods on ML-1M below. The performance boost is still significant.
>
> Our method:  Recall@10=0.116;
>
> MultiVAE [a]: Recall@10=0.101;
>
> LightGCN [b]: Recall@10=0.0987;
>
> DiffRec  [c]: Recall@10=0.106;
>
> On the relational link prediction task, we have already included 13 baselines. We add one more model here.
> HittER (EMNLP 2021, [d]): FB15K-237 (MRR=.373; HR@1=.279; HR@3= .409; HR@10=.558); WN18RR (MRR=.503; HR@1=.462; HR@3=.516; HR@10=.584);  We can see that our model still outperforms them.
>
> ##### **Apart from performance boosts**
> It is worth noting that performance boost is not trivial given the recent advance in these fields. Here, we'd like highlight the two advantages of the proposed method:
> - **Enhanced expressiveness**: the proposed approach can enhance the representation power of current geometric representation learning, and performance boost is the best indicator.
> - **Flexibility**: the model flexibility is shown in two aspects: (a) the proposed approach is applicable to different domains and applications; (b) each data point has the flexibility to choose from a number of candidate product spaces to represent itself (e.g,. Figure 4 and 5).
>
> ##### **Perform some ablation studies to examine the role of l_1 and l_2**
> We conduct ablation studies on the WN18RR dataset.
> - w/o $\ell_1$ and $\ell_2$, we have MRR: 0.515 | H@1: 0.463| H@3: 0.536| H@10: 0.595
> - w/o $\ell_2$ but with $\ell_1$ : MRR: 0.519 | H@1: 0.471 | H@3: 0.540 | H@10: 0.602
> - w/o $\ell_1$ but with $\ell_2$: MRR: 0.520 | H@1: 0.473 | H@3: 0.543| H@10: 0.605
>
> ##### **why geometric representation learning and the potential of the proposed approach**
> Geometric representation learning is fundamental problem and useful in main domains (as reaffirmed by other reviewers). It can introduce a strong inductive bias to the model, making the learning process easier and boosting the expressiveness of the representation.  In the future,
> we plan to carry forward this concept into more sophisticated deep neural networks, such as hyperbolic neural networks or spherical neural networks. In doing so, more complex tasks can benefit from the proposed data-informed geometric space selection idea.
>
> ##### Reference
> - [a] Liang, Dawen, et al. "Variational autoencoders for collaborative filtering."
> - [b] He, Xiangnan, et al. "Lightgcn: Simplifying and powering graph convolution network for recommendation."
> - [c] Wang, Wenjie, et al. "Diffusion Recommender Model." SIGIR (2023).
> - [d] Chen, Sanxing, et al. "HittER: Hierarchical Transformers for Knowledge Graph Embeddings." Proceedings of the 2021 Conference on Empirical Methods in Natural Language Processing. 2021.

---

> > ### Comment · Reviewer_c7Xk · 2023-08-16
> >
> > Thank you very much for the response. It has addressed a good amount of my concerns, but I still have some on experiments.  I mentioned that it lacks comparisons with STOA on matrix completion and link prediction, which means to compare with the STOA results on those two tasks, not just limited to embedding techniques. Such comparison helps assess the practical value of the algorithm.  The added results are still within the embedding field. So I will raise my rating, but one grade up.

---

> > > ### Author Response · Authors · 2023-08-20
> > >
> > > We appreciate that you raise your rating and the opportunity to clarify our work. We want to emphasize that the methods we've listed are not limited to conventional embedding techniques
> > >
> > > For instance, the method DiffRec is recently introduced in SIGIR 2023 (July 23–27, 2023, later than the NeurIPS submission deadline). It is a diffusion model based approach and claims to be the state of the art.  LightGCN is a graph neural network based approach. MultiVAE is a variational autoencoder approach.
> > >
> > > The compared baselines on the relational link prediction task are also not limited to embedding techniques. For instance, ConvE is a convolutional network based approach; M2GNN is a graph neural network based method. HittER is a transformer based method. Here we list two more new methods (although it is not a fair comparison):
> > > - KGTuner [A] (ACL 2022): FB15K-237 (MRR=.352; HR@1=.263; HR@3=.387; HR@10=.530); WN18RR (MRR=0.484; HR@1=0.440; HR@3=.506; HR@10=0.562); This method is based on exhaustive hyper-parameter search. Our method outperform it on all metrics.
> > > - CSProm-KG [B] (ACL 2023, available since 4th July 2023, later than the NeurIPS submission deadline): FB15K-237 (MRR=.358; HR@1=.269; HR@3=.393; HR@10=.538); WN18RR (MRR=0.575; HR@1=0.522; HR@3=.596; HR@10=0.678). This is a concurrent work which uses pretrained large language model (LLM, BERT-Large) as an external source. It is worth noting that WN18RR is sampled from the wordnet lexical database and BERT-Large is trained on a large corpus of lexical data, it is unsurprising that this method can obtain very high score on WN18RR. However, its relatively modest scores on FB15K-237 (sampled from Freebase) suggest limited practicality for non-lexical knowledge graphs.  It is not fair to compare an LLM based approach with our approach as using LLM may lead to data leakage.
> > >
> > > To address any potential misconceptions, our experimental design was primarily aimed at showcasing the superiority of the proposed approach over existing geometric representation learning methods. We hope that this clarification provides a more accurate understanding of our work and its contributions.
> > >
> > > - [A] Yongqi Zhang, Zhanke Zhou, Quanming Yao, and Yong Li. 2022b. Efficient hyper-parameter search for knowledge graph embedding. ACL 2022.
> > > - [B] Chen, Chen, et al. "Dipping PLMs Sauce: Bridging Structure and Text for Effective Knowledge Graph Completion via Conditional Soft Prompting." Findings of the Association for Computational Linguistics: ACL 2023. 2023.

---

> > > > ### Comment · Reviewer_c7Xk · 2023-08-20
> > > >
> > > > Thank you for the response and effort. It clarifies and further addresses my concerns I have further raised my rating.

---

### Official Review · Reviewer_TAMD · 2023-07-04

**Soundness:** 3 good
**Presentation:** 3 good
**Contribution:** 3 good
**Rating:** 7
**Confidence:** 3

**Summary:**

The goal of this paper is to learn the geometry (manifold) underlying given data points. Rather than learning an arbitrary Riemannian manifold from the data, the paper models this manifold as a Cartesian product of manifolds with constant curvature (three prototyprs are use: Euclidean, spherical, hyperbolic). It imposes a certain vector space structure (‘gyrovector’) on these prototypes, where the operations are functions of the curvature. This imposition is a key step as it allows the definition of log and exp maps from the manifolds to tangent spaces, and reach vector representations of points on manifolds.

With this geometry, the paper derives a framework to fit a manifold to the training data. Each data point has components in some number of these manifolds. The paper first estimates the probability of each data point having a component in a specific component manifold. They then add a task specific objective function seeks to optimization data representation in terms of maximizing task performance. With these cost functions, the manifold optimization becomes an optimization problem.

The conceptual applications of this framework include matrix completion, and link prediction for relational graphs. There are a number of experimental results demonstrating the ideas and their performance over the past ideas.


**Strengths:**


-- Improvement over the previous data-driven manifold learning by including curved manifolds as basic components.

-- Interesting use of the gyrovector machinery to derive Euclidean representations for feeding into optimization tools.

-- The experiments provide evidence of success in learning some geometry from the data.



**Weaknesses:**


-- Can one represent arbitrary manifolds using a direct product of these prototypes? Perhaps not. Then what is lost in posing the problem in this way as opposed to a completely nonparametric approach of manifold learning?

-- The paper can provide some intuition on the overall objective function and some more details on how the optimization is performed.




**Questions:**


Minor point:
1.	I was just checking the Mobius sum for elements of a unit sphere in R^n – the result was (Y - <X,Y> X)/(1 - <X,Y>), which is not on the unit sphere. Does it mean that the sum leaves the set {\cal M}_c? Please clarify. This is probably my calculation mistake.

2. Also this expression does not seem to be symmetric in X and Y. Is that correct ?

**Limitations:**


The paper discusses one limitation -- addition of a hyperparameter (K) - -with respect to the previous work on Euclidean product space. However they don't discuss how general their formulation is with respect to an arbitrary data manifold. That is, a manifold with varying curvature.

---

> ### Author Rebuttal · Authors · 2023-08-10
>
> Thank you very much for the positive rating and constructive suggestions!
>
> ##### **Can one represent arbitrary manifolds using a direct product of these prototypes?**
> We cannot represent arbitrary manifolds in this way. We mainly focus on three popular manifolds (spherical, hyperbolic, and euclidean) which have well-defined projections forms and metrics. Other manifolds such as symplectic manifold are out of the scope in this paper.
>
> It is nontrivial to represent manifold with varying curvature in a uniform form. However, the curvature can be learnable in our model. That is, it can be treated as a hyper-parameter or being a learnable parameter that can be optimized together with the model.
>
> ##### **Then what is lost in posing the problem in this way as opposed to a completely nonparametric approach of manifold learning?**
> This study centers around parametric learning algorithms, and the highlighted applications also exhibit a preference for parametric approaches over non-parametric ones. To illustrate, numerous works [a, b] in the literature have shown that matrix factorization yields better results compared to non-parametric techniques like KNN-based models. In the future, we plan to carry forward this concept into more sophisticated deep neural networks, such as hyperbolic neural networks or spherical neural networks.
>
> #### **The paper can provide some intuition on the overall objective function and some more details on how the optimization is performed.**
> Thank you very much for the suggestion, we will add more intuition/explanations on the objective functions to the paper.
>
> First, the two regularization losses can be viewed as a soft constraints on the space selector.
>
> - In the first regularization loss (eq-12), the batch-wise sum of the gate values are considered as the importance of an expert. Minimizing the coefficient of variation the importance distribution encourages all experts to have equal importance.
>
> - The second loss (eq-13) is used to encourage all experts to receive roughly equal numbers of training examples. Although one can ensure equal expert importance with the first loss, the number of training examples received by each expert can be different (e.g., one expert receives a few examples with large weights, and the other receives many examples with small weights). Here, $\kappa$ can be viewed as a soft estimator of the number of examples assigned to each expert for a batch of inputs. Minimizing the coefficient of variation this distribution will help encourage an equal number of training examples for each expert.
>
> Second, the two task specific losses are contrastive losses which encourage positive pairs to be closer and make the distance between negative entities larger.
>
> For the optimization, since all the tensors are projected via the stereographic projection, we use Adam as the optimizer.
>
> ##### **I was just checking the Mobius sum for elements of a unit sphere in R^n – the result was (Y - <X,Y> X)/(1 - <X,Y>), which is not on the unit sphere. Does it mean that the sum leaves the set {\cal M}_c? Please clarify. This is probably my calculation mistake.**
>
> The calculation is correct. However, the mobius sum is defined for the stereographic projection model instead of the original manifold. We provided some examples in Figure 1 and Figure 2 on what the projected model looks like for both hyperbolic and spherical models. We also provided two examples in Figure 3 on the mobius sum calculation results when c=1 and c=-1. See [pdf](https://openreview.net/attachment?id=VzCnW8Uuls&name=pdf).
>
> ##### **Also this expression does not seem to be symmetric in X and Y. Is that correct ?**
> Yes, the expression is not symmetric. But in some special cases, they are symmetric: (1) zero vector case; e.g., of the vector is zero. (2) zero curvature case that is same as Euclidean addition.
>
>
>
> ##### Reference
>
> - [a] Hu, Yifan, Yehuda Koren, and Chris Volinsky. "Collaborative filtering for implicit feedback datasets." 2008 Eighth IEEE international conference on data mining. Ieee, 2008.
> - [b] Koren, Yehuda. "Factorization meets the neighborhood: a multifaceted collaborative filtering model." Proceedings of the 14th ACM SIGKDD international conference on Knowledge discovery and data mining. 2008.

---

### Official Review · Reviewer_Wewy · 2023-07-10

**Soundness:** 3 good
**Presentation:** 3 good
**Contribution:** 2 fair
**Rating:** 5
**Confidence:** 4

**Summary:**

Data representation is an important problem in today's deep learning world. Representation beyond Euclidean geometry, such as spherical or hyperbolic spaces can provide additional flexibility and benefits in capturing underlying properties of data. For example, hyperbolic space can better capture data that has inherent hierarchical structure, and spherical space can better model cyclical structure. In this paper, the authors provide a method to automatically map the individual data points to different geometric spaces using a mixture of experts network. By mapping the data points to different geometric spaces automatically, the authors show advantages in many real world tasks such as matrix completion and link prediction for relational graphs.

**Strengths:**

1) The paper addresses an important problem of automatically mapping the data points to different geometric spaces and motivates the problem well in the paper.

2) This is a challenging problem and many prior methods typically focus on mapping all the data points to a single geometric space.  The formalism is also elegant and the proposed solution shows improvement on matrix completion and graph link prediction over other baselines.

**Weaknesses:**

1) While the matrix completion and link prediction seems like good applications, it is not clear whether the proposed techniques can be extended to some of the other mainstream learning applications in vision and language domains. Furthermore, the early layers in many deep neural networks can already exploit the underlying intrinsic data properties to extract features to show benefits. It is not completely clear whether explicit assignment to individual spaces would provide any additional benefits in some of these newer applications.

2) There has been many prior methods addressing this problem, and the novelty is not explicitly discussed. It would be good if the authors could better clarify this w.r.t MOE and other methods that exploit hybrid geometrical spaces.

3) In many problem settings the underlying data distribution may be predominantly hierarchical or cyclical, and not sure whether it provides strong advantages with the hybrid approaches, and furthermore, the number of geometrical spaces and identification of these geometrical spaces still seem manual and the modern deep learning machinery may be implicitly learning and mapping them as they find them useful in reducing the training loss.

**Questions:**

Overall, this is a well written paper addressing an important problem. Please see concerns above.

**Limitations:**

No concerns.

---

> ### Author Rebuttal · Authors · 2023-08-10
>
> Thank you very much for the positive feedback and reconfirming the importance of the explored problem.
>
> ##### **Q1**: While the matrix completion and link prediction seems like good applications, it is not clear whether the proposed techniques can be extended to some of the other mainstream learning applications in vision and language domains. Furthermore, the early layers in many deep neural networks can already exploit the underlying intrinsic data properties to extract features to show benefits. It is not completely clear whether explicit assignment to individual spaces would provide any additional benefits in some of these newer applications.
>
> **Answer**: Due to a limited rebuttal time frame, we are not able to conduct experiments on the mentioned applications. But we can provide some thoughts on how to apply this idea on these applications. For example, in current mainstream NLP tasks  (e.g., classification, recognizing textual entailment), hyperbolic neural networks [a] (it’s been shown that hyperbolic neural networks can outperform euclidean neural networks[a]), spherical neural works, and normal euclidean neural networks can be built for specific NLP tasks. Then, we can use the sentence representations obtained from pretrained language models (e.g., bert) and the selection network to choose which type of geometric networks to be employed. In doing so, we can also realize the goal of data-informed geometric space selection.
>
>
> ##### **Q2**: There has been many prior methods addressing this problem, and the novelty is not explicitly discussed. It would be good if the authors could better clarify this w.r.t MOE and other methods that exploit hybrid geometrical spaces.
>
> Thank you very much for the constructive suggestion. We will explicitly clarify the relationship of the proposed approach with MOE and  existing hybrid geometrical spaces.
>
> The proposed approach is orthogonal to the existing MOE works. Existing works usually apply the MOE paradigm to euclidean neural networks for better performance or model size scaling up. However, to the best of knowledge, this work is the first attempt to seamlessly integrate the sparsely gated MOE learning paradigm into geometric representation learning. The selected spaces are tightly coupled to form a product space that has strict mathematical definition and meanings.  Compared with existing hybrid geometric spaces (e.g., product space representation learning), in our approach, each data point can inform the model on which geometric space to use, while existing methods treat every data point equally without any customization. In doing so, the proposed method can fully elicit the expressive power of geometric representation learning.
>
>
> ##### **Q3**: In many problem settings the underlying data distribution may be predominantly hierarchical or cyclical, and not sure whether it provides strong advantages with the hybrid approaches, and furthermore, the number of geometrical spaces and identification of these geometrical spaces still seem manual and the modern deep learning machinery may be implicitly learning and mapping them as they find them useful in reducing the training loss.
>
> **Answer**: It is true that in some problem settings, the underlying data distribution can be predominantly hierarchical or cyclical. However, we would like to clarify that we did not make any assumption on the underlying data structure. And in most real-world applications, the underlying data structure is difficult to know. Moreover, our approach subsumes non-hybrid approaches. If we know that the data is predominantly hierarchical, we can set all the candidate space to be hyperbolic.
>
> Pertaining to modern deep learning methods, although it is not in the scope of this paper, we can see from existing literature that non-Euclidean neural networks usually performs better than euclidean neural networks on many tasks [a], indicating that it is not trivial for modern deep learning methods to implicitly inferring the methods. Our method introduces an inductive bias that ensures an effective capture of the data-geometric space relationships, while existing deep learning methods can hardly achieve this.
>
>
>
> ##### Reference
> - [a] Ganea, Octavian, Gary Bécigneul, and Thomas Hofmann. "Hyperbolic neural networks." Advances in neural information processing systems 31 (2018).

---

> ### Comment · Reviewer_Wewy · 2023-08-21
> **Acknowledging the rebuttal**
>
> I thank the authors for the rebuttal. It addresses my concerns and I would like to keep my positive rating.

---

### Official Review · Reviewer_fz8R · 2023-07-27

**Soundness:** 2 fair
**Presentation:** 2 fair
**Contribution:** 3 good
**Rating:** 5
**Confidence:** 3

**Summary:**

In many applications (especially those involving discrete data structures), choosing the right geometry for the embedding space, matching the structure of the data, can lead to significant performance gains. Extant approaches often make an ad hoc choice or use heuristics for the type of geometry applicable globally for the entire data. The paper proposes a novel strategy for the local selection of the product space with appropriate geometry for each data point, using a sparse gating mechanism. The approach is validated on the matrix completion for the movie ratings prediction problem, and link prediction on relational graphs (WordNet, Freebase KG) demonstrating significant performance gains.

**Strengths:**

Strengths of the paper are listed below:

**Relevance** The paper addresses a problem – automatic, and local selection of a product of subspaces with appropriate geometry – of relevance to a wide audience including those working on knowledge graphs, and recommendation systems.

**Originality** While the components of the proposed approach, embedding in different geometric spaces, and sparsely gated MOEs, are not novel, I have not seen the utilization of the latter to make a local (per data-point) selection of the optimal product space. While I’m not deeply familiar with the literature in this area, I believe the proposed approach is novel. I rate the core technical novelty incremental.

**Technical Quality** The technical approach mostly appears sound apart from some doubts that I point to in the weaknesses section.

**Evaluation** The evaluation using the tasks of (a) matrix completion in the movie-ratings setting, and (b) link prediction in relational graphs (WordNet, and Freebase KG), and the demonstrated significant gains in performance (especially on link prediction in the Freebase KG over the SOTA) validate the utility of the proposed approach. This strength is weakened by points listed in the weaknesses section.

**Significance** The approach has the potential for significant impact over wide areas of research once the weaknesses have been addressed.

**Weaknesses:**

Weaknesses of the paper are listed below:

**Technical Quality** I'm not convinced regarding the appropriateness of using the CNN layers $f_1(.)$ and $f_2(.)$. Since the vectors $e_p^{(i)}$ lie in different spaces, even if lifted to the corresponding tangent spaces, $f_j(.)$ layer would perform algebra on them and take weighted combinations of entities (embeddings) in entirely different spaces. To me this doesn’t seem proper.

**Evaluation**
- Matrix completion for movie-ratings can be considered a classical problem with a lot of algorithms benchmarked on MovieLens 1M. However, the paper doesn’t directly provide comparisons with the state of the art (bar chart in Figure 3). Secondly, it doesn’t provide a table with quantitative results, enabling a better understanding of the exact performance gains achieved.
- Figure 6 shows that while the performance may be more robust to the choice of N, K has a significant impact. In the spirit of local selection of the relevant product spaces, it stands to reason that optimal K may also vary from point to point. Was this aspect investigated?

**Clarity**
- (l. 111) What is a ‘Rheinmain Geometric Space’? If a typo, kindly fix, else provide a definition with a reference to the literature.
- (l. 201-208; (12)-(15)) Kindly add explanations for how the two regularizers, $l_1$ and $l_2$,  achieve load balancing. Point to the relevant literature where they were introduced or mention that they are novel introductions.
- (Fig. 6) The value of K for the left panel, and N for the right panel are not identified.
- some typos need to be fixed (not considered in evaluating the paper).

**Questions:**

Kindly address the weaknesses pointed in the critique above. I summarize them below:
- appropriateness of algebra involving embeddings in different spaces.
- quantitative comparison with SOTA on the movie-ratings problem
- local selection of K.
- Clarify with references (a) Rheinmain geometric space, (b) how the regularizers achieve load balancing.
- Discuss limitations of the work (see below)

**Limitations:**

Authors don’t address limitations of the work.

I don’t think there are any direct negative societal implications. Other limitations and opportunities for improvement are addressed in my responses to previous questions.

---

> ### Author Rebuttal · Authors · 2023-08-10
>
> Thank you very much for confirming the novelty as well as the potential of the proposed approach. Also, the suggestions can certainly help us improve the manuscript.
>
> Below we answer the questions and make some clarifications.
>
> ##### **Why do we use CNN layers (appropriateness of algebra involving embeddings in different spaces)?**
> All the vectors are initialized in the Euclidean space then mapped to corresponding geometric spaces, the CNN layers are applied over the initialized euclidean vectors regardless of which geometric space these vectors will be mapped to for simplicity. We agree that it might be possible to use hyperbolic/spherical neural networks in the process. However, in this case, it requires each geometric space to have its own selection network, and we still need to combine the outputs of all the selection networks to arrive at a single decision. We will leave it as a future work to explore more advanced selection mechanisms.
>
> ##### **Comparison with state of the art on Movielens**
> We provide comparisons with other stoa methods on ML-1M below. We can see that, compared with the most recent models, the performance gain is still significant.
>
> Our method:  Recall@10=0.116;
>
> MultiVAE (WWW 2018 [a]), an variational autoencoder based model; Recall@10=0.101;
>
> LightGCN (SIGIR 2020 [b]), a graph convolutional network based model: Recall@10=0.0987;
>
> DiffRec (SIGIR 2023, first available on April 2023 [c[), a diffusion recommender model; Recall@10=0.106;
>
> ##### **local selection of K**
> In our current design, K is selected for the whole dataset instead of each data point. It is reasonable that each data point may require a different K, but it is quite challenging to achieve this goal. One potential solution is to build an MOE network with different K for each data point, but this can be unrealistic as the model size increases linearly to the data size.
>
> Moreover, K is usually set to a small value in our experiments (select from 2, 3, 4), as such, manual hyper-parameter search is good in this case.
>
> ##### **Clarification**
> - (1) Rheinmain geometric space ⇒ it is a typo, and it should be Riemannian geometric space.
> - (2) The loading balance approach is not a novel contribution, but from reference ***. We will add these references [d][e] to the paper.
>
> The two regularization losses can be viewed as a soft constraints on the space selector.
> - In the first regularization loss (eq-12), the batch-wise sum of the gate values are considered as the importance of an expert. Minimizing the coefficient of variation the importance distribution encourages all experts to have equal importance.
> - The second loss (eq-13) is used to encourage all experts to receive roughly equal numbers of training examples. Although one can ensure equal expert importance with the first loss, the number of training examples received by each expert can be different (e.g., one expert receives a few examples with large weights, and the other receives many examples with small weights). Here, $\kappa$ can be viewed as a soft estimator of the number of examples assigned to each expert for a batch of inputs. Minimizing the coefficient of variation this distribution will help encourage an equal number of training examples for each expert.
>
>
> ##### **Discuss limitations of the work**
> We have a short subsection in 3.4 to discuss the limitations of the proposed method. Based on the review, we also identified one more limitations: the proposed approach can not assign different K for each data point.
>
> ##### **Reference**
> - [a] Liang, Dawen, et al. "Variational autoencoders for collaborative filtering." Proceedings of the 2018 world wide web conference. 2018.
> - [b] He, Xiangnan, et al. "Lightgcn: Simplifying and powering graph convolution network for recommendation." Proceedings of the 43rd International ACM SIGIR conference on research and development in Information Retrieval. 2020.
> - [c] Wang, Wenjie, et al. "Diffusion Recommender Model." SIGIR (2023).
> - [d] Shazeer, Noam, et al. "Outrageously large neural networks: The sparsely-gated mixture-of-experts layer." arXiv preprint arXiv:1701.06538 (2017).
> - [e] Bengio, Emmanuel, et al. "Conditional computation in neural networks for faster models." arXiv preprint arXiv:1511.06297 (2015).

---

> > ### Comment · Reviewer_fz8R · 2023-08-20
> > **Thanks for the response.**
> >
> > Thanks for the response. I don't have any further questions.

---

### Author Rebuttal · Authors · 2023-08-10

We thank all the reviewers for their suggestions and we answer corresponding questions under each review's rebuttal. The pdf contains some figures to show the stereographic projection models for hyperbolic & spherical spaces. Mobius sum is also demonstrated in this pdf.

---

### Decision · Program_Chairs · 2023-09-21

**Decision:**

Accept (poster)

**Comment:**

The authors propose a method to embed data in constant curvature manifolds (i.e., Euclidean, hyperbolic, and spherical spaces) and their product manifolds. In doing so, they propose a mechanism to adaptively and sparsely select product manifolds, using a gating mechanism. The efficacy of the proposed method is verified on several problems, including matrix completion.  This paper was well received by reviewers, acknowledging the clarity of the manuscript and the value of the core idea. During the authors and reviewers' discussion period, the authors provide detailed responses to all raised concerns. As such, the AC agrees with the evaluation by the reviewers and recommends the acceptance of the work; congratulations. Please revise your work based on comments and incorporate the new results provided during the rebuttal phase.